# On the Scalability of Certified Adversarial Robustness with Generated Data

**Thomas Altstidl**[1]    **David Dobre**[3]    **Arthur Kosmala**[5]    **Björn Eskofier**[1,2]
**Gauthier Gidel**[3,4]    **Leo Schwinn**[5]

[1] Machine Learning and Data Analytics Lab, FAU Erlangen Nürnberg, Germany
[2] Institute of AI for Health, Helmholtz Zentrum München, Germany
[3] Mila, Université de Montréal, Canada    [4] Canada CIFAR AI Chair
[5] Data Analytics and Machine Learning, Technische Universität München, Germany

{thomas.r.altstidl,bjoern.eskofier}@fau.de
{david-a.dobre,gidelgau}@mila.quebec  {a.kosmala,l.schwinn}@tum.de

## Abstract

Certified defenses against adversarial attacks offer formal guarantees on the robustness of a model, making them more reliable than empirical methods such as adversarial training, whose effectiveness is often later reduced by unseen attacks. Still, the limited certified robustness that is currently achievable has been a bottleneck for their practical adoption. Gowal et al. and Wang et al. have shown that generating additional training data using state-of-the-art diffusion models can considerably improve the robustness of adversarial training. In this work, we demonstrate that a similar approach can substantially improve deterministic certified defenses but also reveal notable differences in the scaling behavior between certified and empirical methods. In addition, we provide a list of recommendations to scale the robustness of certified training approaches. Our approach achieves state-of-the-art deterministic robustness certificates on CIFAR-10 for the $\ell_2$ ($\epsilon = 36/255$) and $\ell_\infty$ ($\epsilon = 8/255$) threat models, outperforming the previous results by $+3.95$ and $+1.39$ percentage points, respectively. Furthermore, we report similar improvements for CIFAR-100.

## 1  Introduction

Deep learning models have been successfully applied for a variety of different applications. However, it is widely recognized that the vulnerability of neural networks to adversarial examples [1] remains an open problem and hinders their adoption in safety-critical domains. Prior research on improving the robustness of neural networks against adversarial examples can be broadly classified into empirical [2, 3] and certified approaches [4].

Adversarial training is currently the most prominent empirical robustification method [3]. Here, the training data of neural networks is augmented with adversarial examples, improving the robustness against attacks at inference time. Recent work has demonstrated that adversarial training can be considerably improved using synthetically generated data, even without training the generative model with external data [5, 6]. Nevertheless, empirical robustness has repeatedly been shown to be ineffective against more sophisticated attacks developed in subsequent work [7].

In contrast to empirical methods, certified approaches yield robustness guarantees given a predefined threat model, most often based on the $\ell_1$, $\ell_2$, or $\ell_\infty$ norm. As a result, these methods provide reliable protection against future attacks. Nevertheless, the robustness guarantees achieved by certification

38th Conference on Neural Information Processing Systems (NeurIPS 2024).

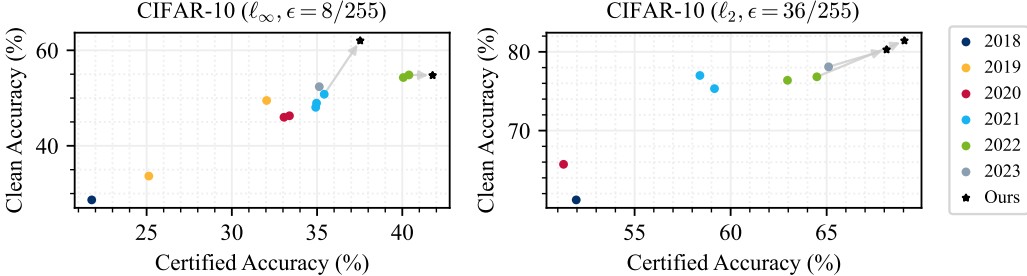

Figure 1: Certified and clean accuracy of top-ranked models on CIFAR-10 taken from the SoK Certified Robustness for Deep Neural Networks [10] leaderboard. By using data generated by an elucidating diffusion model (EDM), accuracy significantly improves for four different models and two different norms ($\ell_\infty$ and $\ell_2$). Grey arrows indicate improvements stemming from this data augmentation.

methods are generally substantially lower than the robustness obtained by empirical defenses for the same threat model [3, 4, 8].

In this work, we aim to analyze how well certified robustness scales when utilizing additional data generated by diffusion models during the model training, a recipe that has previously proven successful for empirical robustness. Only Hu et al. [9] have already used generated data, and thus a broader in-depth review of the factors influencing scalability is to date missing. In our empirical study, we analyze models trained to be robust against $\ell_\infty$ and models trained to be robust against $\ell_2$ norm attacks. The proposed approach improves robustness for both the $(\ell_\infty, \epsilon = 8/255)$ and $(\ell_2, \epsilon = 36/255)$ threat models on CIFAR-10, improving upon the previous results in the literature by $3.95\%p$ and $1.39\%p$ (percentage points), respectively. In most experiments, the increase in certified accuracy is accompanied by an increase in accuracy on clean data, where we observe improvements by up to $4.83\%p$. Figure 1 summarizes the improvements compared to the previous state-of-the-art with respect to clean and certified accuracy on CIFAR-10. Further experiments show that the same approach considerably improves certified accuracy on CIFAR-100 as well.

Moreover, we conduct ablations to evaluate the impact of different design choices, including regularization, the number of training epochs, the optimization schedule, and the optimal balance between real and generated data. We summarize the most important insights of this empirical study in a list of recommendations that can be followed to more accurately compare and improve the robustness of deterministic certified defenses. Lastly, we discern crucial differences in the scaling behavior between empirical and certified methods. All code used to produce the results and figures in this paper will be released on GitHub after publication.

## 2   Related Work

**Empirical Robustness.**   Adversarial training was first introduced by Goodfellow et al. [2]. The authors employed the single-step Fast Gradient Sign Method (FGSM) to craft adversarial examples during training and thereby robustify the model against these attacks. Later research by Madry et al. [3] demonstrated that single-step adversarial training does not yield considerable robustness against multi-step attacks. They showed that using the multi-step Projected Gradient Descent (PGD) attack during training successfully improves the robustness of neural networks at test time, even against strong attacks. Subsequent work proposed improvements to the loss function, the adversarial attack used during training, and better trade-offs between clean and certified accuracy [11, 12].

**Certified Robustness.**   Unlike empirical methods, certified methods yield robustness guarantees, thereby eliminating possible vulnerabilities to future attacks. Certification methods can be broadly classified into two methodologically distinct groups, namely probabilistic and deterministic methods. Probabilistic methods aim to approximate smooth classifiers using Monte Carlo sampling and noise injection [4]. A given sample is verified as robust with a certain probability depending on the noise magnitude and number of Monte Carlo samples. To obtain a tight verification bound, probabilistic methods need to perform a substantial amount of sampling procedures (forward passes)

for each sample, considerably increasing the computational overhead in practice. Contrary to probabilistic methods, deterministic approaches do not entail considerable computational overhead during inference.

Convex bound propagation [13, 14, 15, 16] is a group of deterministic methods that leverages interval arithmetic and linear programming to track how perturbations in the input space transform through each layer, effectively constructing an outer envelope that contains all possible network outputs for inputs within the specified perturbation region. Our initial results scaling the work by Palma et al. [16], provided in Appendix H, show that this yields severely deteriorated performance when training with additional generated data and is thus a poor candidate for scaling certified robustness.

Our main focus is hence devoted to deterministic approaches that bound the Lipschitz constant of each neural network layer to be small (generally smaller or equal to 1) for a predefined $\ell_p$ norm [17, 18]. The Lipschitz constant of the whole network is bounded by the multiplication of the Lipschitz constants of the individual layers [19]. Given a network's upper bound of the Lipschitz constant, a robustness guarantee can be trivially obtained by computing the distance between the highest two logits in the output space.

**Diffusion Models.** More recently, diffusion models have superseded generative adversarial networks (GANs) as the preferred method for image generation [20]. Denoising diffusion probabilistic models (DDPM) [21] can generate high-quality samples on CIFAR-10 [22] with an FID score of 3.17, a common measure of image quality. Since then, other variants have been proposed [23, 24]. By further analyzing the design space of these models [23], elucidating diffusion models (EDMs) achieve a current state-of-the-art FID score of 1.79 on CIFAR-10. With additional discriminator guidance [24], the quality of these EDM-generated images are reported to reach an FID score of 1.64, the best score reported in literature for CIFAR-10 at the time of writing.

**Improving Empirical Robustness with Auxiliary Data.** Hendrycks et al. [25] showed that utilizing additional data from external datasets during adversarial training can improve empirical adversarial robustness. Gowal et al. [5] extended this approach to synthetically generated data from generative models only trained on the source dataset. Recently, Wang et al. [6] showed that leveraging the latest advances in diffusion models further improves empirical adversarial robustness.

In this work, we investigate if leveraging data generated with state-of-the-art diffusion models can also improve certified robustness against adversarial attacks and analyze how certified training approaches can be scaled optimally.

## 3 Experiment Setup

Given the recent improvements in adversarial training using additional data generated by diffusion models, we devise a set of experiments to investigate whether this also transfers to certified robustness. We focus on deterministic methods as probabilistic methods entail a tremendous computational overhead during inference time and do not achieve considerable robustness for the $\ell_\infty$ norm yet [10]. All our experiments are done on a single Nvidia A100 graphics card (40GB of VRAM) without distributed training.

### 3.1 Dataset and Threat Models

We perform experiments on CIFAR-10 and CIFAR-100 [22], for which EDM-generated data is readily available and a wealth of previous robustness research exists [6]. Our experiments and ablation studies focus on CIFAR-10. We refrain from experiments on larger datasets like ImageNet [26] as robustness guarantees achieved by deterministic methods for these datasets are still comparatively low, and only [9] support it at the time of writing. We perform experiments on two common threat models, specifically $(\ell_\infty, \epsilon = 8/255)$ and $(\ell_2, \epsilon = 36/255)$ adversaries. We do not consider the $\ell_1$ threat model, as only smoothing-based approaches achieve considerable robustness for this threat model at the time of writing. For our experiments, we select the two best architectures from the popular certified robustness leaderboard introduced by Li et al. [10] for both the $\ell_2$ (GloroNet [9] and LOT [17]) and $\ell_\infty$ threat models (SortNet [18] and $\ell_\infty$-dist Net [27]). In total, we perform experiments on architectures from four different papers.

Table 1: Clean and certified test accuracy (%) on **CIFAR-10** ($\ell_\infty, \epsilon = 8/255$) for $\ell_\infty$-dist Net and SortNet with dropout rate $\rho$. Bold highlights the best model with and without auxiliary data. $|\mathcal{D}_{gen}|$ denotes the number of EDM-generated images at which highest accuracy was achieved.

| Net | Epochs | w/o auxiliary | | w/ auxiliary | | |
| --- | --- | --- | --- | --- | --- | --- |
| | | Clean | Cert. | Clean | Cert. | $|\mathcal{D}_{gen}|$ |
| $\ell_\infty$-dist | 800 | 57.34 | 34.25 | 61.04 | 36.98 | 1m |
| | 1600 | 57.19 | 34.00 | 62.02 | 37.53 | 1m |
| SortNet | 3000 | 53.38 | **39.72** | 53.29 | 41.32 | 10m |
| $\rho = .85$ | 6000 | 53.36 | 39.05 | 52.41 | 40.70 | 1m |
| SortNet | 3000 | 56.09 | 37.44 | 54.36 | 41.71 | 5m |
| $\rho = .00$ | 6000 | 54.81 | 36.50 | 54.75 | **41.78** | 10m |

## 3.2 Generated Auxiliary Data

To explore the effectiveness of augmenting the original CIFAR-10 and CIFAR-100 [22] datasets with generated data, we adjust the data loader of each model to use a fraction of generated data and original data in every epoch. We use the same generated data used by Wang et al. [6], which was produced by an EDM trained only on the train set of CIFAR-10. In a preliminary experiment, we found the generated-to-real ratio to be optimal when 30% of training images are real and 70% are generated in every epoch during training, matching the ratio used by Wang et al. [6]. We performed experiments with $50,000$ (50k), $100,000$ (100k), $200,000$ (200k), $500,000$ (500k), 1 million (1m), 5 million (5m) and 10 million (10m) generated images. Wang et al. [6] sub-sampled the 1m images from 5m images choosing only the $20\%$ most confidently classified images according to a pretrained WRN-28-10 model. In contrast, we naively sub-sample the 1m images from the 5m image dataset to avoid potential selection bias by the classifier used to select the data. Moreover, using the same selection process for all datasets should allow us to assess better the effect of the amount of generated data on the final robustness.

## 3.3 Hyperparameters

With additional data, it is also expected that both model size and the number of training epochs can be further scaled to improve clean accuracy and robustness. We thus perform experiments on the influence of model depth and the number of epochs on clean and certified accuracy. For some models, we investigate further techniques that add learning capacity. Concretely, for SortNet [18] we also experiment with models that do not employ dropout, and for LOT [17] we adjust the learning rate scheduler to cosine annealing [28].

## 4 Results

In the following, we first summarize the effect of using additional generated data on the achievable certified and clean accuracy. Furthermore, we ablate the effect of other design choices on the certified robustness, such as the number of training epochs, model size, the amount of additional synthetic data and other hyperparameters. Lastly, we summarize our findings and provide a list of recommendations to scale certified robustness effectively.

### 4.1 Improving Certification Approaches with Generated Data

Across all four reference models and all two threat models, we find that the inclusion of generated data can improve certified accuracy. In most cases, clean accuracy is considerably improved as well. An overview of our new state-of-the-art results in comparison with existing related work is given in Figure 1. For the ($\ell_\infty, \epsilon = 8/255$) threat model on CIFAR-10 we can increase the robustness of the existing SortNet [18] to $41.78\%$, an improvement of $1.39$ percentage points. For the ($\ell_2, \epsilon = 36/255$) model we achieve a certified accuracy of $69.05\%$ using LOT [17], a substantial increase of $3.95$ points compared to the best result previously reported in the literature [10]. In almost all cases this

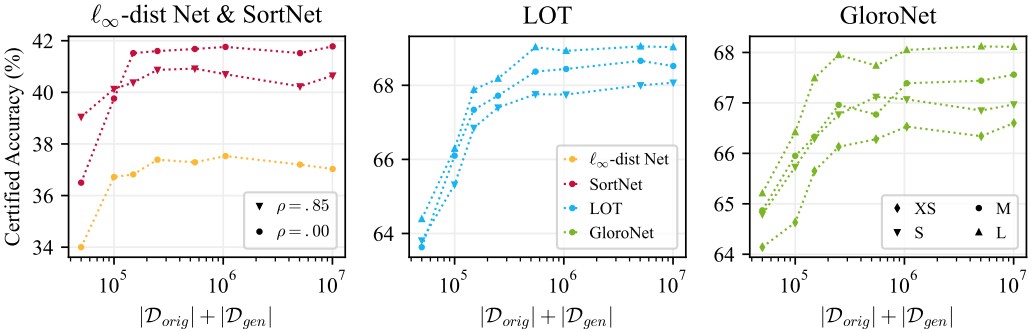

Figure 2: Influence of total CIFAR-10 dataset size $|\mathcal{D}_{orig}| + |\mathcal{D}_{gen}|$ (number of original and generated images) on certified accuracy. $\rho$ is the dropout rate of SortNet. All models were trained with a large number of epochs, i.e., 1600 for $\ell_\infty$-dist Net, 6000 for SortNet, 600 for LOT, and 2400 for GloroNet. Accuracy generally improves little beyond 1m generated images.

improvement coincides with an increase in clean accuracy. One exception is SortNet, where the clean accuracy slightly decreases from $54.84\%$ to $54.75\%$.

Full results are given in Tab. 1 for both SortNet and $\ell_\infty$-dist Net, as well as Tab. 2 for LOT and Tab. 3 for GloroNet. We find that by removing the dropout from SortNet we can improve certified accuracy from $41.32\%$ to $41.78\%$ when using auxiliary data. However, the same leads to a drop from $39.72\%$ to $37.44\%$ with only the original data, reinforcing the notion that the additional data acts as a good regularizer. Similarly, for LOT, cosine annealing is superior by a margin of up to $1.12$ points compared to a multi-step scheduler, indicating that the model can make better use of its capacity when trained with auxiliary data. For full results using the multi-step scheduler, we refer to App. B, Tab. 2 – all results discussed in subsequent sections refer to those obtained with cosine annealing.

Adding auxiliary data improves certified robustness on CIFAR-100 as well, as demonstrated in Tab. 4. We restrict our evaluation to the most effective models from the analysis on CIFAR-10 and do not further scale the number of training epochs. The most substantial improvements on CIFAR-100 are obtained for the SortNet model, where certified accuracy increases by $8.08$ percentage points from $9.2\%$ to $17.28\%$, and for GloroNet, where certified accuracy increases by $2.49$ percentage points from $36.41\%$ to $38.9\%$.

## 4.2 Sensitivity Analysis

**Scaling the Amount of Auxiliary Data.** The main goal of this work was to evaluate the influence of additional synthetic data during training on the achievable certified accuracy of deterministic certification methods. To this end, we analyze how different amounts of additional data affect the final certified robustness. We evaluate the saturation by training with with 50k, 100k, 200k, 500k, 1m, 5m, and 10m auxiliary generated images. All models are trained for at least twice the amount of epochs that were used in the original papers to ensure saturation in terms of training time. As seen in Fig. 2, improvements in certified robustness beyond 1m are mostly negligible. This behavior is also largely independent of model size, which stands in contrast to prior results reported for adversarial training [5].

**Scaling the Model Size.** We performed experiments on several different model sizes to investigate possible correlations between the benefit of additional training data and the model capacity. The $\ell_\infty$-based models are largely constructed out of fully connected layers. As a result, the computational effort when scaling these models increases quadratically. As experiments on the $\ell_\infty$-based models proved to be computationally too expensive we refrain from scaling these models and focus instead on $\ell_2$-based models.

Tables 1 to 3 demonstrate that scaling the model size can increase the certified robustness for both LOT and GloroNet. Shown are the best improvements when adding 1m, 5m, or 10m auxiliary data when compared to the same model trained without any auxiliary data – referred to as the *base model* from here on. The highest gains are for medium models with LOT and for large models with

Table 2: Clean and certified test accuracy (%) on **CIFAR-10** ($\ell_2, \epsilon = 36/255$) for LOT. Bold highlights the best model with and without auxiliary data. $|\mathcal{D}_{gen}|$ denotes the number of EDM-generated images at which highest accuracy was achieved.

| Size | Epochs | w/o auxiliary | | w/ auxiliary | | |
|------|--------|-------|-------|-------|-------|-----------------------|
|      |        | Clean | Cert. | Clean | Cert. | $|\mathcal{D}_{gen}|$ |
| S    | 200    | 76.60 | 63.45 | 79.22 | 66.17 | 10m |
|      | 400    | 76.50 | 63.48 | 80.02 | 67.42 | 5m  |
|      | 600    | 76.75 | 63.81 | 80.59 | 68.07 | 10m |
| M    | 200    | 76.92 | 63.38 | 79.59 | 66.95 | 5m  |
|      | 400    | 76.41 | 63.93 | 80.53 | 68.17 | 5m  |
|      | 600    | 76.49 | 63.63 | 80.98 | 68.66 | 5m  |
| L    | 200    | 77.21 | **64.53** | 79.91 | 67.58 | 10m |
|      | 400    | 77.00 | 64.47 | 80.80 | 68.66 | 5m  |
|      | 600    | 76.62 | 64.39 | 81.42 | **69.05** | 5m  |

Table 3: Clean and certified test accuracy (%) on **CIFAR-10** ($\ell_2, \epsilon = 36/255$) for GloroNet. Bold highlights the best model with and without auxiliary data. $|\mathcal{D}_{gen}|$ denotes the number of EDM-generated images at which highest accuracy was achieved.

| Size | Epochs | w/o auxiliary | | w/ auxiliary | | |
|------|--------|-------|-------|-------|-------|-----------------------|
|      |        | Clean | Cert. | Clean | Cert. | $|\mathcal{D}_{gen}|$ |
| XS   | 800    | 76.51 | 63.44 | 77.42 | 65.57 | 5m  |
|      | 1600   | 76.95 | 63.79 | 78.38 | 66.48 | 10m |
|      | 2400   | 77.56 | 64.14 | 78.54 | 66.60 | 10m |
| S    | 800    | 77.22 | 64.33 | 77.90 | 66.23 | 10m |
|      | 1600   | 77.91 | 64.68 | 78.74 | 66.78 | 1m  |
|      | 2400   | 78.28 | 64.79 | 78.89 | 67.07 | 1m  |
| M    | 800    | 77.73 | 64.81 | 77.95 | 66.59 | 5m  |
|      | 1600   | 77.77 | 65.06 | 79.18 | 67.31 | 10m |
|      | 2400   | 78.41 | 64.87 | 79.43 | 67.56 | 10m |
| L    | 800    | 77.94 | 65.09 | 79.13 | 67.06 | 10m |
|      | 1600   | 78.99 | 65.16 | 79.81 | 67.67 | 1m  |
|      | 2400   | 79.33 | **65.21** | 80.28 | **68.12** | 5m  |

GloroNet. For GloroNet in particular we observe that model size becomes more important the longer the model is trained, with all models trained for the default 800 epochs showing similar gains.

**Scaling the Number Training Epochs.** As larger models and additional training data may require longer model training to achieve optimal results we increased the number of training epochs compared to the original configurations for all tested models[1]. For the $\ell_\infty$-based models in Tab. 1 we see that by doubling the number of epochs we can further improve certified robustness when regularization is removed. A similar picture arises for the $\ell_2$-based models, presented in Tabs. 2 and 3. Here, we find that an increase in the number of epochs yields a significant improvement regardless of model size. Overall, this parameter had the strongest impact in combination with auxiliary data and all best certified accuracies are achieved at their respective maximum number of epochs, with the only exception being SortNet with dropout.

## 4.3 Relationship between Generalization Gap and Certified Robustness

The individual improvements in certified robustness vary considerably between different model and data configurations in our experiments. One possible explanation for these differences may be that the generalization gap of the respective models trained without auxiliary data – i.e., the *base models* – are different, leading to different gains when closing this generalization gap. To investigate this, we

---

[1]During training of CIFAR-10 a model will see $0.7 \cdot 50{,}000 = 35{,}000$ generated images in each epoch.

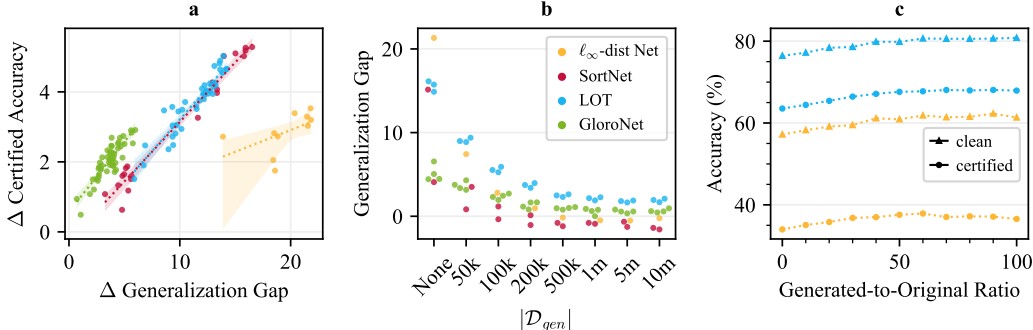

Figure 3: **a** Correlation between generalization gap and certified accuracy improvement. Generalization gap measures difference between training and testing accuracy. $\Delta$ refers to difference between base model and model trained with auxiliary data. **b** Generalization gaps by amount of auxiliary data $|\mathcal{D}_{gen}|$ for all models trained with maximal epoch count. **c** Clean and certified accuracy (%) for different ratios of generated and real data for $\ell_\infty$-dist Net and LOT-S. Here, a generated-to-original ratio of 70 means 70% of each batch is generated data and the remaining 30% is real data.

correlate the difference in generalization gap with the improvement in certified accuracy obtained when adding auxiliary data. Here, generalization gap refers to the difference between the train and test accuracy on clean data for the best epoch. Figure 3a demonstrates a considerable correlation between the decrease in generalization gap compared to the base model trained with no auxiliary data and the improvement in certified robustness for models with auxiliary data. Moreover, we perform a line fit between the generalization gap and robustness improvement for all the analyzed models. Surprisingly the slope of the different lines is similar for most models, except $\ell_\infty$-dist Net, indicating that robustness gains can be predicted once the offset of the line is known for unseen models. However, the offset of the different lines depends on the base model considered. Generalization gaps for each auxiliary dataset size $|\mathcal{D}_{gen}|$ in Fig. 3b also correlate well with scaling curves in Fig. 2.

These results are in line with certified robustness gains achieved for SortNet with and without dropout shown in Tab. 1. Here, the certified robustness of the SortNet model trained without auxiliary data and for 3000 epochs decreases when using less regularization by removing dropout from 39.72% to 37.44%. At the same time, the generalization gap of the two models increases from 4.16% to 12.89%, respectively, as an effect of removing dropout. However, once additional synthetic data is used, removing dropout actually improves the certified robustness to up to 41.71% by nearly two points. Here, increasing the generalization gap by removing dropout had a positive effect on the final certified robustness, which fits observations in Fig. 3a.

### 4.4 Ratio of Generated and Real Data

In every training epoch, we use a proportion of synthetic and real images and keep the total amount of images the same as the size of the original training set. The default configuration throughout our experiments, and the one also used by Wang et al. [6], is to use 30% real images and 70% generated images in each batch. Figure 3c illustrates how using different proportions for generated and real data affects the certified robustness of the $\ell_\infty$-dist Net and LOT-S architectures. We see that at ratios of 60% generated data the clean accuracy saturates, and with 70% the certified accuracy saturates. Notably, in all cases the accuracy when only training with generated images was higher than when only training with real images, indicating that it may be possible to fully train these models on only generated data in the future.

### 4.5 Certification Radius Distribution

To examine the underlying factors contributing to the observed increase in robustness when using auxiliary data, we conduct an analysis of the certification radius distribution for the SortNet (without dropout) and LOT-L architectures. Figure 4 displays the number of images on the $y$-axis with a certification radius equal to or above a specific value, shown on the $x$-axis. Curves are plotted for the

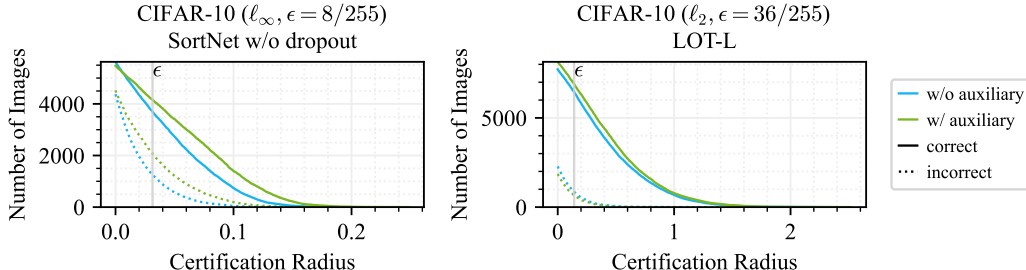

Figure 4: Cumulative distribution of certification radii for the best $\ell_\infty$-based model, Sortnet w/o dropout, and the best $\ell_2$-based model, LOT-L. Note how for SortNet images exhibit overall smaller certification radii for both correct and incorrect classes, yet clean accuracy slightly decreases.

best models trained with and without auxiliary data. Additionally, curves for correct and incorrect classifications are displayed separately.

We observe no considerable differences between the distribution of certification radii for the LOT model obtained with or without auxiliary data. Nevertheless, models trained with auxiliary data show slightly higher robustness for correctly classified samples and lower robustness for misclassified samples on average. The SortNet architecture exhibits considerably higher robustness radii for both correct and incorrect classifications when using auxiliary data. Here, differences in certified robustness do not seem to come from a better generalization ability on clean data but from larger certification radii on unseen data. On the other hand, the LOT architecture shows similar certification radii but considerably better generalization on clean data. A more detailed analysis is given in App. G, Fig. 3. Both models show considerable over-robustness for a considerable fraction of the test set, where the certification radius is well beyond the certification goal $\epsilon$.

### 4.6 Takeaways for Scaling Certified Robustness

Differences in certified robustness between distinct defense approaches are often marginal and even small improvements over prior work may be relevant. Here, we summarize the most important takeaways from the empirical study presented in this work on how to scale the robustness of deterministic certified models.

- **Scale the number of training epochs.** Among all investigated hyperparameters, we found the number of training epochs had the most consistent effect on certified accuracy when training with auxiliary data. Based on our experiments, we expect the amount of auxiliary data to not matter as long as it is sufficient for closing the generalization gap.

- **Increase your model capacity.** When using auxiliary data, a large generalization gap between training and testing accuracies is less of an issue and, based on our results, indicative of untapped performance improvements that can be leveraged (see Sec. 4.3). This means model capacity can be scaled with little fear of overfitting. Our experiments show that reducing the amount of regularization, using better optimizers, and increasing the model size all improve certified accuracy when using auxiliary data without leading to large generalization gaps.

- **Compare with caution.** Even small adjustments, such as a change to dropout, learning rate schedulers, or even different random seeds can improve certified accuracy by about $0.5$ to $1$ percentage points. This makes it difficult to assess the individual contribution of different architectures towards robustness – for example, contrary to results reported in the original papers, our results indicate that LOT may actually be superior to GloroNet.

- **Benchmark with generated auxiliary data.** As the proposed approach does not entail a computational overhead for the same amount of training epochs, we recommend future work to compare their approaches using auxiliary data and ensure that models are trained till convergence. Differentiating between approaches that use auxiliary data and those that only utilize the original dataset may be helpful for future benchmarks. Similar approaches have been adopted in the empirical robustness domain [29].

- **Decrease over-robustness.** While not related to the usage of auxiliary data, our evaluation in Fig. 4 indicates that a considerable number of samples are noticeably more resistant to adversarial attacks than what was intended during training. Future research may consider using smaller certification objectives for samples that already demonstrate considerable robustness, an approach that has already been shown to be successful in adversarial training [12].

## 5 Comparison to Empirical Robustness

While we observe many of the same trends as Gowal et al. [5] and Wang et al. [6], i.e., that larger models and more epochs generally help, we also note a few crucial differences.

- **Amount of training data.** Scaling beyond one million CIFAR-10 generated images did not further improve certified accuracy, regardless of model size. This is different to scaling behavior reported previously for adversarial training, with, e.g., Gowal et al. [5] reporting an improvement of $+2.65$ and $+2.52\%$ when scaling their WRN-23-10 and WRN-70-16 models, respectively, from one million to 100 million generated images. One hypothesis would be that this is due to the consistency property of the Glivenko-Cantelli class, mentioned by Béthune et al. in Section 5.1 [30], due to which Lipschitz-1 neural networks' training loss converges to the testing loss for increasingly large datasets. In contrast, adversarial training can be interpreted as a data-dependent operator norm regularization [31] and seems to require more data samples to close the generalization gap. More recent work suggests that the Bayes error may limit the achievable accuracy of both deterministic [32] and probabilistic [33] certified robustness, in line with our results.

- **Overfitting between best and last epoch.** In adversarial training, prior work finds that the difference between the best and the last epoch becomes increasingly smaller with larger amounts of auxiliary data [6]. In contrast, we observe no such effect for certifiably robust models (see App. E, Fig. 1), highlighting another difference in scaling behavior between empirical and certified robustness.

- **Optimal generated-to-original ratio.** Previous research regarding the optimal generated-to-original ratio for adversarial training observes a drop-off when using only generated data. Both Gowal et al. [5] (Fig. 5) and Wang et al. [6] (App. B, Fig. 3) suggest accuracy decreases again significantly beyond 70% generated data when using diffusion models. As evident in Fig. 3c this is not as pronounced for certifiably robust models.

Together, these results suggest that for certifiably robust models it may be possible to determine a sufficient amount of generated data beforehand for any given dataset - contrary to what is the case for adversarial training, where more is always better. Moreover, it indicates that certified robustness is considerably harder to scale than empirical robustness, as once saturation with respect to data has been achieved, further gains can only be attained by better algorithms and increased model size. Future experiments should explore concrete scaling laws for certified and empirical adversarial robustness, which was out-of-scope of this paper. Deriving and verifying theoretical properties of certifiably robust models, such as those already known on graphs [34], also remains an exciting topic for further research.

Table 4: Clean and certified test accuracy (%) on **CIFAR-100** for both ($\ell_\infty, \epsilon = 8/255$) and ($\ell_2, \epsilon = 36/255$) threat models. Bold highlights the best overall model for each architecture.

| Architecture | w/o auxiliary | | w/ auxiliary | | |
|---|---|---|---|---|---|
| | Clean | Cert. | Clean | Cert. | $|\mathcal{D}_{gen}|$ |
| $\ell_\infty$-dist Net | 25.99 | 9.36 | 27.73 | **10.47** | 10m |
| SortNet $\rho = .00$ | 24.93 | 9.20 | 27.58 | **17.28** | 1m |
| LOT-L | 46.60 | 32.92 | 50.68 | **36.56** | 5m |
| GloroNet-L | 51.57 | 36.41 | 51.71 | **38.90** | 10m |

# 6 Limitations

Despite our best efforts and extensive experiments, some limitations remain. Our focus is largely on deterministic Lipschitz-bound methods, which incidentally were used in the two best-performing models for the $(\ell_\infty, \epsilon = 8/255)$ and $(\ell_2, \epsilon = 36/255)$ attacks on CIFAR-10 at the time the experiments were performed [10]. Other approaches for certified robustness, including probabilistic ones and other deterministic methods, stay largely unexplored except for our experiments on convex bound propagation in Appendix H. The inherently high computational complexity of the evaluated models [9, 17, 18, 27] also inhibited us from running each configuration multiple times, as even a single run already results in an overall compute time of around five thousand GPU hours. Finally, we made the conscious choice to not perform experiments on ImageNet as only one of the evaluated models, that by Hu et al. [9], supported this dataset and thus no meaningful comparison could be made.

# 7 Conclusion

We show that deterministic certified robustness can be improved by up to $5.28\%p$ when additional generated data from a diffusion model is used during training. This is true across four different architectures and two different threat models, $(\ell_\infty, \epsilon = 8/255)$ and $(\ell_2, \epsilon = 36/255)$, on CIFAR-10, where we report improved certified accuracies of $41.78\%$ and $69.05\%$, respectively. For $\ell_\infty$ we are thus able to achieve a new state-of-the-art, while Hu et al. [9] report a slightly better accuracy of $70.1\%$ for the $\ell_2$ threat model using data generated by DDPM. In addition, we show the data scaling also improves certified accuracy on CIFAR-100 substantially.

We find that the highest gains can be achieved for models where the generalization gap, i.e., the difference between training and testing accuracy, is high for the original model. When augmenting with generated data, the generalization gap is mostly eliminated across all models when a sufficient amount of additional images are used. As the generalization gap gets smaller, removing regularization techniques, such as dropout, and switching to learning rate schedulers aimed at better convergence yields additional improvements. We also note that increasing the number of epochs had the greatest impact when paired with generated data. Lastly, we observe that a considerable number of samples are noticeably more resistant to adversarial attacks than required by the $\epsilon$-bound.

## Acknowledgments

Thomas Altstidl acknowledges the support by the Bavarian State Ministry of Health and Care, project grant number PBN-MGP-2010-0004-DigiOnko. Leo Schwinn gratefully acknowledges funding by the Deutsche Forschungsgemeinschaft (DFG, German Research Foundation) - Projectnumber 544579844.

## Author Contributions

T.A. and L.S. came up with the original idea of using generated data for certified robustness. T.A. implemented the code adjustments, ran the experiments, performed the analysis, prepared all figures/tables, and wrote the manuscript. D.D. supported some experiments with additional compute. All authors contributed to the interpretation of the results and to the editing of the manuscript.

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

## A  Model Details

Table 5 lists the different model configurations used. We limit our summary to the most important parameters – for the full parameters used during training we refer to the respective papers [9, 11, 17, 18] and repositories, as well as our own scripts provided in our code release.

Table 5: Model Configurations

| Model | Configuration | Depth | Width | Comments | Code |
|---|---|---|---|---|---|
| $\ell_\infty$-dist Net | – | 6 | 5120 | | |
| SortNet | $\rho = .85$ | 6 | 5120 | | GitHub |
| SortNet | $\rho = .00$ | 6 | 5120 | | |
| LOT | S | 10 (2 blocks) | n/a | LipConvnet-10 | |
| LOT | M | 20 (4 blocks) | n/a | LipConvnet-20 | GitHub |
| LOT | L | 40 (8 blocks) | n/a | LipConvnet-40 | |
| GloroNet | XS | 6 | 128 | LiResNet L6W128 | |
| GloroNet | S | 12 | 128 | LiResNet L12W128 | |
| GloroNet | M | 18 | 128 | LiResNet L18W128 | GitHub |
| GloroNet | L | 18 | 256 | LiResNet L18W256 | |

## B  Results LOT Multi-Step Scheduler

Table 6 summarizes the results of LOT trained with the multi-step learning rate scheduler used in the original paper. Overall the results are approximately $0.5\%$ worse than using a cyclic learning rate.

Table 6: LOT w/ Multi-Step Scheduler, $\ell_2$, $\epsilon = 36/255$

| Architecture | Epochs | No | | 1m | | 5m | | 10m | |
|---|---|---|---|---|---|---|---|---|---|
| | | Clean | Cert. | Clean | Cert. | Clean | Cert. | Clean | Cert. |
| LOT-S | 200 | 75.64 | 62.43 | 78.69 | 65.06 | 78.40 | 65.01 | 78.82 | 65.09 |
| | 400 | 75.74 | 62.95 | 79.58 | 66.36 | 79.40 | 66.51 | 79.33 | 66.41 |
| | 600 | 75.85 | 63.02 | 80.03 | 66.97 | 80.04 | 66.97 | 79.86 | 66.95 |
| LOT-M | 200 | 77.03 | 63.60 | 79.14 | 66.30 | 79.30 | 66.40 | 79.29 | 66.12 |
| | 400 | 76.83 | 63.42 | 80.01 | 67.37 | 80.35 | 67.67 | 80.15 | 67.61 |
| | 600 | 76.47 | 63.56 | 80.53 | 67.98 | 80.54 | 68.19 | 80.17 | 67.82 |
| LOT-L | 200 | 76.90 | 63.70 | 79.35 | 66.60 | 79.33 | 66.79 | 79.53 | 66.66 |
| | 400 | 76.76 | 64.23 | 80.42 | 67.84 | 80.52 | 67.78 | 80.21 | 67.80 |
| | 600 | 76.93 | 64.35 | 80.73 | 68.70 | 81.08 | 68.40 | 80.59 | 68.54 |

# C   Full Results 1m/5m/10m

Table 7: Clean and certified test accuracy (%) on **CIFAR-10** ($\ell_\infty, \epsilon = 8/255$) for $\ell_\infty$-dist Net and SortNet. Bold and italics highlight the best model with and without auxiliary data and underlining highlights the overall best model. $\Delta$Cert. denotes the highest absolute increase in certified robustness when using auxiliary data, and $e$ the number of epochs.

| Architecture | $e$ | No | | 1m | | 5m | | 10m | | $\Delta$Cert. |
|---|---|---|---|---|---|---|---|---|---|---|
| | | Clean | Cert. | Clean | Cert. | Clean | Cert. | Clean | Cert. | |
| $\ell_\infty$-dist Net | 800 | 57.34 | *34.25* | 61.04 | **36.98** | 60.35 | 36.00 | 60.64 | 36.30 | +2.73 |
| | 1600 | 57.19 | 34.00 | 62.02 | **37.53** | 61.39 | 37.20 | 61.40 | 37.03 | +3.53 |
| SortNet | 3000 | 53.38 | *39.72* | 52.50 | 41.23 | 52.78 | 40.35 | 53.29 | **41.32** | +1.60 |
| w/ dropout | 6000 | 53.36 | 39.05 | 52.41 | **40.70** | 52.57 | 40.23 | 53.09 | 40.65 | +1.65 |
| SortNet | 3000 | 56.09 | *37.44* | 54.28 | 41.51 | 54.36 | **41.71** | 54.18 | 41.41 | +4.27 |
| w/o dropout | 6000 | 54.81 | 36.50 | 54.72 | 41.76 | 54.36 | 41.52 | 54.75 | **41.78** | +5.28 |

Table 8: Clean and certified test accuracy (%) on **CIFAR-10** ($\ell_2, \epsilon = 36/255$) for LOT and GloroNet. Bold and italics highlight the best model with and without auxiliary data and underlining highlights the overall best model. $\Delta$Cert. denotes the highest absolute increase in certified robustness when using auxiliary data, $e$ the number of epochs, and XS, S, M, and L the model size.

| Architecture | | $e$ | No | | 1m | | 5m | | 10m | | $\Delta$Cert. |
|---|---|---|---|---|---|---|---|---|---|---|---|
| | | | Clean | Cert. | Clean | Cert. | Clean | Cert. | Clean | Cert. | |
| LOT | S | 200 | 76.60 | 63.45 | 79.19 | 65.81 | 79.18 | 65.93 | 79.22 | **66.17** | +2.72 |
| | | 400 | 76.50 | 63.48 | 79.87 | 66.99 | 80.02 | **67.42** | 79.96 | 67.23 | +3.94 |
| | | 600 | 76.75 | *63.81* | 80.36 | 67.75 | 80.56 | 68.00 | 80.59 | **68.07** | +4.26 |
| | M | 200 | 76.92 | 63.38 | 79.58 | 66.85 | 79.59 | **66.95** | 79.90 | 66.82 | +3.57 |
| | | 400 | 76.41 | *63.93* | 80.37 | 67.96 | 80.53 | **68.17** | 80.75 | 68.08 | +4.24 |
| | | 600 | 76.49 | 63.63 | 80.71 | 68.44 | 80.98 | **68.66** | 80.69 | 68.52 | +5.03 |
| | L | 200 | 77.21 | *64.53* | 80.06 | 67.34 | 80.29 | 67.47 | 79.91 | **67.58** | +3.05 |
| | | 400 | 77.00 | 64.47 | 80.80 | 68.39 | 80.80 | **68.66** | 80.76 | 68.51 | +4.19 |
| | | 600 | 76.62 | 64.39 | 81.24 | 68.93 | 81.42 | **69.05** | 81.20 | 69.03 | +4.66 |
| GloroNet | XS | 800 | 76.51 | 63.44 | 77.30 | 65.29 | 77.42 | **65.57** | 77.45 | 65.17 | +2.13 |
| | | 1600 | 76.95 | 63.79 | 78.13 | 66.24 | 78.21 | 65.98 | 78.38 | **66.48** | +2.69 |
| | | 2400 | 77.56 | *64.14* | 78.98 | 66.53 | 78.34 | 66.34 | 78.54 | **66.60** | +2.46 |
| | S | 800 | 77.22 | 64.33 | 78.12 | 66.17 | 77.99 | 66.14 | 77.90 | **66.23** | +1.90 |
| | | 1600 | 77.91 | 64.68 | 78.74 | **66.78** | 78.44 | 66.62 | 78.53 | 66.59 | +2.10 |
| | | 2400 | 78.28 | *64.79* | 78.89 | **67.07** | 78.76 | 66.85 | 78.87 | 66.97 | +2.28 |
| | M | 800 | 77.73 | 64.81 | 78.11 | 66.37 | 77.95 | **66.59** | 78.02 | 66.45 | +1.78 |
| | | 1600 | 77.77 | *65.06* | 78.83 | 67.12 | 79.04 | 66.83 | 79.18 | **67.31** | +2.25 |
| | | 2400 | 78.41 | 64.87 | 79.57 | 67.39 | 79.44 | 67.44 | 79.43 | **67.56** | +2.69 |
| | L | 800 | 77.94 | 65.09 | 79.12 | 66.97 | 78.94 | 66.80 | 79.13 | **67.06** | +1.97 |
| | | 1600 | 78.99 | 65.16 | 79.81 | **67.67** | 79.50 | 67.28 | 79.90 | 67.60 | +2.51 |
| | | 2400 | 79.33 | *65.21* | 80.20 | 68.05 | 80.28 | **68.12** | 80.00 | 68.11 | +2.91 |

## D Full Results CIFAR-100

Table 9: Clean and certified test accuracy (%) on **CIFAR-100** for both $(\ell_\infty, \epsilon = 8/255)$ and $(\ell_2, \epsilon = 36/255)$ threat models. Bold highlights the best overall model for each architecture.

| Architecture | No | | 1m | | 5m | | 10m | | $\Delta$Cert. |
|---|---|---|---|---|---|---|---|---|---|
| | Clean | Cert. | Clean | Cert. | Clean | Cert. | Clean | Cert. | |
| $\ell_\infty$-dist Net | 25.99 | 9.36 | 27.56 | 10.45 | 27.69 | 10.25 | 27.73 | **10.47** | +1.11 |
| SortNet w/o dropout | 24.93 | 9.20 | 27.58 | **17.28** | 26.74 | 16.69 | 27.50 | 17.25 | +8.08 |
| LOT-L | 46.60 | 32.92 | 50.52 | 36.50 | 50.68 | **36.56** | 50.59 | 36.22 | +3.64 |
| GloroNet-L | 51.57 | 36.41 | 51.78 | 38.54 | 51.81 | 38.66 | 51.71 | **38.90** | +2.49 |

## E Robust Overfitting

In Figure 5 we visualize the number of epochs between the epoch where the best-certified accuracy was achieved and the total amount of epochs trained on the $x$-axis. On the $y$-axis we plot the certified accuracy difference between the best and last epoch. Here, the auxiliary data used for the different configurations are visualized with unique colors and symbols. No clear connection between the amount of auxiliary data and the distance between the best and last epoch can be observed. However, the highest distance is observed for models using no auxiliary data. On average, the observed difference between the last and best epoch is small for all models and always below $0.5\%$. We conclude that robust overfitting is not an issue for the certified training approaches tested in our experiments.

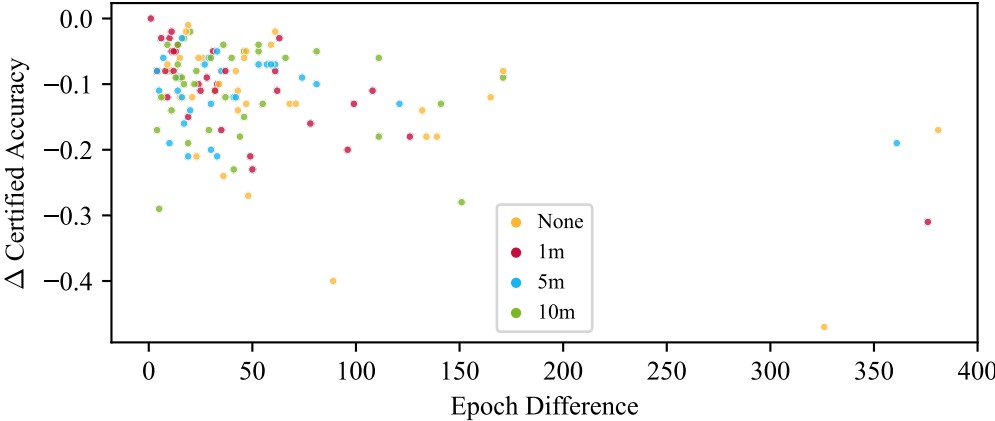

Figure 5: Relationship between 1) the amount of auxiliary data used, 2) the number of epochs between the epoch where the best certified accuracy was achieved, and 3) the difference in certified accuracy between the best and last epoch.

# F    Influence Model Size/Epoch Count

Figure 6 visualizes the effects we also mention in our sensitivity analysis in Sec. 4.2. Looking at each row, increasing epoch counts yields to a linear increase in improvements, with the sole exception of GloroNet-XS. Model size similarly has a positive effect on certified accuracy, especially in combination with increases of epoch counts.

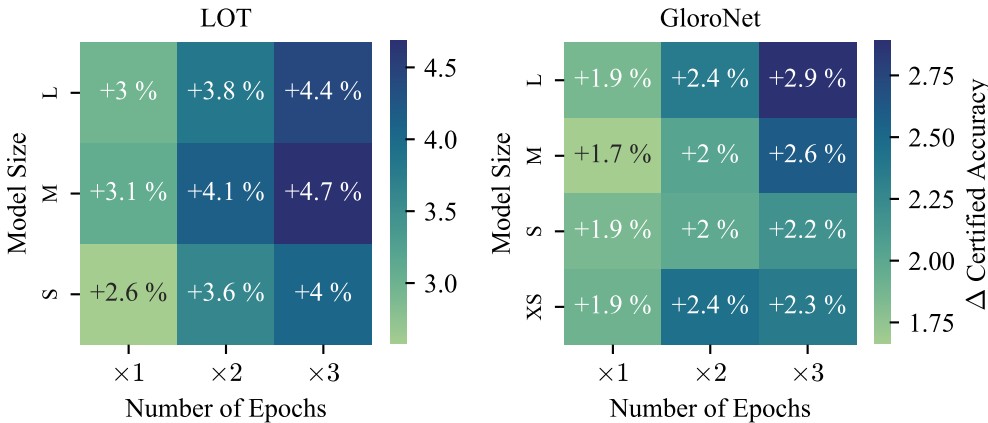

Figure 6: Influence of model size and increase in the number of epochs for CIFAR-10 ($\ell_2, \epsilon = 36/255$) models. Color shading indicates average absolute improvement across 1m, 5m, and 10m auxiliary data over the same model trained without auxiliary data.

# G    Certification

Figure 7 illustrates the difference between the best base model (w/o auxiliary) and the best model trained with auxiliary data (w/ auxiliary). We investigate different combinations of correctness and certification for each image. An image may either be correctly or incorrectly classified, and either certified or not certified. If it is certified, this means that its certification radius is larger than $\epsilon$.

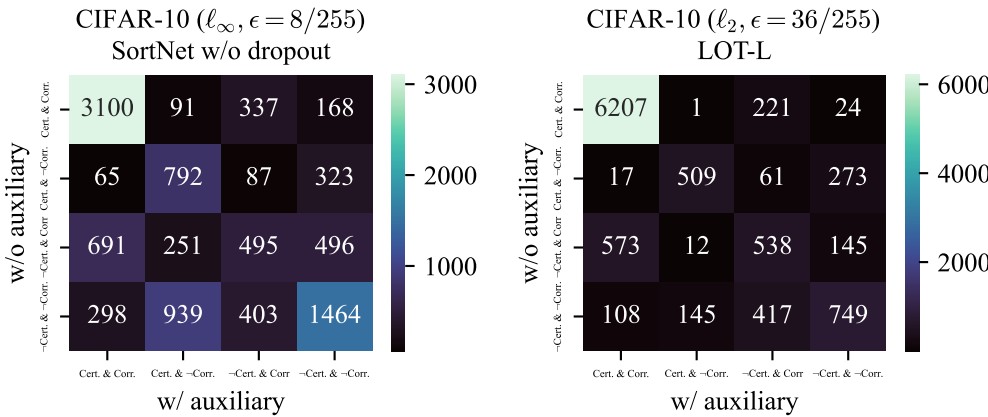

Figure 7: Confusion matrices for different correctness and certification constellations between models trained with and without auxiliary data. Here, ¬ means not, i.e., ¬Cert. means that images were not certified to be within the $\epsilon$-bound.

# H  Convex Bound Propagation

Our focus on Lipschitz-bound methods may raise questions regarding the applicability of data scaling to other certified robustness domains, such as convex bound propagation. Our additional experiments on MTL-IBP [16], a summary of which is reproduced below, indicate vastly different scaling dynamics with no observed improvements even when further adjusting (being originally used by Palma et al. [16] for the larger ImageNet). A few remarks:

- Verification runs were aborted after 24 hours to accommodate the discussion period. If aborted, reported numbers indicate percentage of certified/correctly classified samples on the subset of CIFAR-10 test images for which verification completed within 24 hours.
- Non-completed runs included more hard-to-certify samples that required computationally expensive branch-and-bound verification. Extrapolated full runtimes for some configurations are up to about 2 weeks.
- Runs that yielded 0% certified accuracy oftentimes completed quickly as the comparatively cheap PDG attack already found counterexamples, and thus a full verification was not needed.

We conclude that MTL-IBP, a convex bound propagation approach, cannot benefit from additional generated data. This indicates that results with regards to scalability do not trivially transfer from one robustness method to another, and a detailed investigation of a single method, such as Lipschitz-bound approaches, seems justified.

Table 10: Clean (top) and certified (bottom) test accuracy (%) on **CIFAR-10** ($\ell_\infty, \epsilon = 8/255$) for MTL-IBP. Bold highlights the best overall model for each configuration. $e$ denotes the number of epochs, $\alpha$ the expressive loss coefficient.

| $e$ | $\alpha$ | None | 50k | 100k | 200k | 500k | 1m | 5m | 10m |
|-----|----------|------|-----|------|------|------|-----|-----|------|
| 260 | 0.5 | **35.08%** | 34.14% | 32.04% | 21.07%* | 0.34%* | † | 0.00% | 0.00% |
| 520 | 0.5 | **35.95%** | 34.74% | 34.57% | 33.05% | 17.57%* | 0.54%* | 0.00%* | 0.00% |
| 260 | 0.05 | **28.71%**\* | 26.67%* | 23.75%* | 8.12%* | 0.00%* | † | 0.00% | 0.00% |
| 520 | 0.05 | **27.76%**\* | 27.26%* | 26.68%* | 26.72%* | - | - | - | - |

| $e$ | $\alpha$ | None | 50k | 100k | 200k | 500k | 1m | 5m | 10m |
|-----|----------|------|-----|------|------|------|-----|-----|------|
| 260 | 0.5 | 53.69% | 55.46% | 57.70% | 66.04%* | 79.31%* | † | 94.60% | **94.82%** |
| 520 | 0.5 | 54.68% | 55.64% | 55.88% | 57.34% | 68.63%* | 79.10%* | 94.60%* | **95.09%** |
| 260 | 0.05 | 64.58%* | 66.12%* | 67.67%* | 76.35%* | 84.92%* | † | 94.67% | **94.84%** |
| 520 | 0.05 | 65.94%* | 66.08%* | 66.73%* | **67.77%**\* | - | - | - | - |

\* Verification run did not finish within 24 hours, preliminary result on test data subset shown
† Implementation by Palma et al. [16] triggered an assertion error

