# OpenReview forum: "On the Scalability of Certified Adversarial Robustness with Generated Data"
_NeurIPS.cc/2024/Conference — NeurIPS 2024 poster_

### Official Review · Reviewer_Vssv · 2024-06-20

**Soundness:** 3
**Presentation:** 4
**Contribution:** 3
**Rating:** 5
**Confidence:** 4

**Summary:**

The paper presents an empirical study on how synthetic data can help improve the robustness accuracy and clean accuracy of certified adversarial robustness. While existing studies have shown promising results in using synthetic data to improve empirical adversarial robustness, the effectiveness of synthetic data on certified robustness has never been explored. This paper bridges this gap by designing experiments on $l_2$ and $l_\infty$ robustness of multiple models over the CIFAR-10 and CIFAR-100 datasets. Experiment results show that synthetic data improves certified robustness but in a different way than their effects on the empirical one. The paper also provides ablation studies and guidance on different hyperparameters, such as dropout rate, epochs, model sizes, learning rate schedulers, quantity, and ratios between original and synthetic data.

**Strengths:**

1. The paper conducts experiments with multiple models and two datasets, demonstrating the generalizability of their study.
2. The ablation studies consider several hyperparameters such as dropout rate, epochs, model sizes, learning rate schedulers, quantity, and ratios between original and synthetic data.
3. The paper provides guidance on comparing certified accuracy among different approaches. This guidance is important and should be prompted in this research field.

**Weaknesses:**

1. My biggest concern is that the paper misses a large chunk of related works in section 2 and its experiments.
    * In section 2, when discussing the deterministic approaches of certified robustness, the paper writes, "one deterministic approach consists of ..." There are other groups of deterministic approaches using convex bound propagation, such as IBP, SABR, TAPS [1, 2, 3]. The paper focuses only on approaches bounding the Lipschitz constant of each neural network layer. This narrow focus hurts the generalizability of the paper and should be justified. For example, the paper mentions that this group (bounding the Lipschitz constant) of approaches generates a robust guarantee by computing the distance between the highest two logits in the output space. This sentence justifies why not focus on the convex bound propagation approaches.
    * In experiments, the paper's generalizability can be enhanced by comparing against or combining state-of-the-art convex-bound-propagation approaches, such as SABR and TAPS, SOTA, to see how the synthetic data can help.

2. In section 4.3, the paper discusses the correlation between the generalization gap and certified accuracy. The reasoning in this section makes sense. However, the paper might neglect the fact that the model trained with dropout and synthetic data hurts the certified accuracy. This is a weird result. Could you provide the generalization gap of the model trained with $\rho=0.85$ and synthetic data to try to explain this wired result?

[1] On the Effectiveness of Interval Bound Propagation for Training Verifiably Robust Models

[2] Certified training: Small boxes are all you need

[3] Connecting Certified and Adversarial Training

**Questions:**

1. In section 3.2, the paper mentions that "we naively sub-sample the 1m images from the 5m image dataset to..." Are those images from generated data or original data? Also, the sizes of the generated data are different (50K~5m). Do you always perform a 20% sub-sampling, and where do the constants "1m" and "5m" come from?
2. Does the diffusion model used to generate synthetic data see the robust classifier? Will it generate different synthetic data for different classifiers?
3. In section 5, the paper mentions "overfitting between best and last epoch." Does the paper perform experiments comparing one model trained by early stopping with the other without the early stopping?

Comment:

The paper points out a distinction between the scale of synthetic data on certified robustness and empirical robustness, i.e., scaling beyond one million CIFAR-10 generated images did not further improve certified accuracy, regardless of model size. This observation might be related to Bayes error [4,5].

[4] Certified Robust Accuracy of Neural Networks Are Bounded due to Bayes Errors

[5] How Does Bayes Error Limit Probabilistic Robust Accuracy

**Limitations:**

Although the paper mentions some limitations in their checklists, one significant limitation is whether synthetic data can help other deterministic training methods, such as convex bound propagation, as mentioned in the weakness.

---

> ### Author Rebuttal · Authors · 2024-08-06
>
> Thank you for your thoughtful comments, which we address below.
>
> 1. While we would have loved to include an even more extensive set of models (and datasets), our focus was on the best two models for the ℓ2 and ℓ∞ norms on CIFAR-10. In particular, for CIFAR-10 with ℓ∞, ε=8/255 threat model the IBP has 32.04% accuracy (67.96% error rate), SABR has 35.13%, and TAPS 35.10%. We have, however, now expanded our discussion of related work to include convex bound propagation, and are actively trying to add experiments for one paper of that family of certified methods.
> 2. We are uncertain whether we have correctly interpreted your inquiry, so what follows is our best attempt at resolving any confusion. The best SortNet model w/o auxiliary data has 39.72% certified accuracy, with a generalization gap of 4.00%pt. When removing dropout (going down Tab. 1), the gap increases to 12.55%pt. When adding auxiliary data (going to the right in Tab. 1) the gap decreases to -0.82%pt, i.e., testing accuracy is better than training accuracy, indicating negative effects of dropout. Hence the overall best SortNet model with 41.78% certified accuracy has no dropout, and is able to benefit from longer training. We’d also like to point out that for ρ=0.85, the model w/ auxiliary data (41.32%) still outperforms the one w/o auxiliary data (39.72%), and the bold simply denotes the best model w/o auxiliary data.
>
> Regarding your questions, we answer them as follows.
>
> 1. Wang et al. provide their generated data at https://github.com/wzekai99/DM-Improves-AT, which is also where the 1m, 5m, and 10m constants are from. However, their 1m dataset was subsampled from the 5m using the confidences of a classifier trained on the original CIFAR-10 images. This is why we instead use the first 20% of the generated 5m images.
> 2. No, the diffusion model was conditionally trained on the ground-truth CIFAR-10 labels, but did not at any point use a classifier.
> 3. Appendix E contains a figure comparing the best versus last epochs, and the full result tables in supplementary material directory cert-robust also contain columns with both. We did not find any meaningful correlation, which is also a marked difference compared to adversarial robustness.
>
> We have now included a discussion of the Bayes error in Sec. 5 “Amount of Training Data.” Undoubtedly further exploration is required to determine why this behavior differs, and we remain confident our paper provides the necessary impulse to answer these and related questions.

---

> > ### Comment · Reviewer_Vssv · 2024-08-07
> >
> > Thank you for your response. I realized that I made a mistake when reading Table 1 regarding the second point of weakness. I will keep my score for now and I need to further discuss with other reviewers and ACs about the first point of weakness.

---

> > > ### Author Response · Authors · 2024-08-12
> > >
> > > Thank you for encouraging us to perform additional experiments on convex bound propagation, for which we provide additional results in our overall rebuttal. In short, MTL-IBP performs significantly worse when successively adding auxiliary data and its scaling dynamics deviate significantly from those for Lipschitz-bound models. We'd kindly ask you to consider this in your final evaluation.

---

### Official Review · Reviewer_bWYd · 2024-07-06

**Soundness:** 3
**Presentation:** 3
**Contribution:** 3
**Rating:** 5
**Confidence:** 4

**Summary:**

The paper explores advancements in certified defenses against adversarial attacks in deep learning models by leveraging data from state-of-the-art diffusion models during training. It addresses the current challenges where empirical methods, such as adversarial training, augment data but face difficulties with new attacks, contrasting with certified approaches that provide robustness guarantees within predefined threat models but often exhibit lower overall robustness. By integrating additional data from diffusion models, the study achieves state-of-the-art robustness certifications on CIFAR-10 and CIFAR-100 under L infinite and L2 threat models. This approach not only enhances deterministic defenses but also improves accuracy on clean data. The paper also performs extensive ablation studies to examine the effects of various design choices on certified robustness.

**Strengths:**

1. The writing of the paper is clear and easy to follow.
2. The paper explores a novel approach by leveraging data from diffusion models to improve certified defenses, which are typically more reliable than empirical methods.
3. The approach achieves state-of-the-art results on CIFAR-10, demonstrating a significant improvement in robustness certificates for both L infinite and L2 threat models.
4. The experimental findings are comprehensive and solid.

**Weaknesses:**

1. The paper evaluated four architectures selected from the certified robustness leaderboard [9]. However, the achieved clean accuracy and robustness accuracy on these neural networks are notably low. For instance, compared to architectures like WideResNet, which can attain at least 95% clean accuracy and over 80% robustness accuracy on CIFAR-10, the accuracies achieved by the approach in this paper remain insufficient.
2. While evaluating the impact of using generated data from diffusion models to bolster certified robustness is novel, the idea itself represents a somewhat incremental advance. Previous research has already explored the use of such data to enhance adversarial robustness  within empirical methods.
3. Despite the appeal of certified robustness due to its robustness guarantees, its lower accuracy compared to empirical methods diminishes its practicality. While this paper enhances certified robustness, its results have limited practical applicability.

**Questions:**

1. The clean accuracy and robust accuracy of the networks trained in this paper are still lower than those of other architectures. Could you explain why they are lower and provide some reasons for this?
2. Considering the relatively low accuracies, how practical is certified robustness? Could you list several practical scenarios where sacrificing accuracy for robustness to this extent would be justified?

**Limitations:**

1. The results primarily focus on CIFAR-10 and CIFAR-100 datasets, with limited exploration of generalization to other datasets and real-world scenarios.
2. The results are based on four selected neural network architectures, and the generalization to a broader range of architectures is not thoroughly investigated.

---

> ### Author Rebuttal · Authors · 2024-08-06
>
> Thank you for taking the time to provide valuable feedback. We’d like to address your concerns regarding the weaknesses as follows.
>
> 1. This is currently a well-known general limitation of certified robustness, and we believe should thus not have any influence in how our work is evaluated. To date, there is a trade-off to be made between high levels of robust accuracy to predefined attack models (empirical robustness) or lower levels of accuracy to any theoretically possible attack model (Lipschitz-bound certified robustness, among others), the latter of which is obviously a much stronger guarantee. For what it’s worth, the work by Hu et al. [1] raises hope that some of these limitations may be overcome in future research.
> 2. The idea itself may not be novel, yet its evaluation on a broad set of Lipschitz-bound models certainly is. While incremental, we still believe it is of interest to the wider community focused on certified robustness, as it establishes that a better baseline for comparing certified models may be the data-saturated regime where no generalization gap is present. This better reflects the capability of each model to certify examples, rather than its ability to generalize.
> 3. Practicality is inherently subjective, yet we agree there are limits to certified robustness that prevent wide-spread adoption. However, we would also like to highlight that the direct comparison of adversarial accuracies to certified ones is inherently ill-posed, as the latter is strictly smaller than the former, which is dependent on a specific attack model (e.g., PGD). It is thus unfair to dismiss them on the basis of this comparison alone. While an in-depth discussion of advantages is beyond the scope of a rebuttal, we echo the sentiments of Mangal et al. [2] who postulate that Lipschitz-bound certified defenses may put an end to the cat-and-mouse game of adversarial defenses and attacks.
>
> While some points have already been mentioned as part of our discussion of the weaknesses above, we’d also like to respond to the questions.
> 1. Unlike adversarial robustness, which is always evaluated with a specific attack, certified models present guarantees with respect to all possible attacks (given a norm and epsilon bound). This is achieved by imposing mathematical constraints, such as orthogonality, that have negative impacts on model capacity and computational complexity. Both limitations are currently being addressed in ongoing research.
> 2. While practicality is somewhat limited currently, safety-critical domains remain interested in Lipschitz-bound certified robustness as it is immune to advanced attack models that may be devised by malicious actors in the future as part of the cat-and-mouse game that is partially fueling the large amount of publications in adversarial robustness. We’d again like to point to Mangal et al. [2] who more extensively argue regarding its importance.
>
> [1] K. Hu, A. Zou, Z. Wang, K. Leino, and M. Fredrikson, “Unlocking Deterministic Robustness Certification on ImageNet,” in Adv. Neural Inf. Process. Syst. (NeurIPS), vol. 36, New Orleans, LA, USA, Dec. 2023.
>
> [2] R. Mangal, K. Leino, Z. Wang, K. Hu, W. Yu, C. Pasareanu, A. Datta, and M. Fredrikson, “Is Certifying ℓp Robustness Still Worthwhile?,” Tech. Rep., Oct. 2023, arXiv:2310.09361 [cs] type: article. [Online]. Available: https://arxiv.org/abs/2310.09361

---

> > ### Comment · Reviewer_bWYd · 2024-08-12
> >
> > Thanks for your clarification. My concerns about the incremental nature and the limited practical applicability of this work still remain. I believe these aspects are still critical. I will keep my score.

---

### Official Review · Reviewer_hXmp · 2024-07-13

**Soundness:** 2
**Presentation:** 3
**Contribution:** 2
**Rating:** 6
**Confidence:** 4

**Summary:**

This work proposes using data augmentation with diffusion models to improve the certified robustness of image classification models. The authors analyze the training and certification behavior of different Lipschitz-bound-based machine learning models when the training data is supplemented with additional generated samples.

**Strengths:**

**The Method Improves Certified Robustness**

The experiments demonstrate that adding generated data can enhance the certified robustness of several certification methods. Three out of the four tested methods show improvements when trained on additional data, and the fourth method also shows improvements when dropout is removed. The authors suggest this is due to a generalization gap in existing models, which can be bridged by training on additional (generated) data.

**Extensive Evaluation**

The evaluation section is comprehensive, with tests conducted across two datasets, four certification methods, and two threat models. It also includes numerous ablations, such as varying model sizes and percentages of generated data. This provides a well-rounded understanding of the proposed data generation method, with findings effectively summarized in the takeaway list in Section 4.6.

**Code Included and Well Documented**

The supplementary material includes code for running the experiments, which enhances reproducibility. The repository is well-documented, providing clear instructions for setup and reproduction of the paper’s results, tables, and figures.

**Weaknesses:**

**Limited Novelty and Insights**

The idea of supplementing training data with generated data is not new and has been applied in various contexts. Specifically, it has already been used to improve Lipschitz certification by Hu et al. [22]. While this work offers a more thorough evaluation across different threat models and model architectures, the additional insights gained are limited. The paper mainly serves as an evaluation without providing new theoretical insights or methodological advancements.

**Missing Comparison to a Key Related Approach**

A very similar work by Hu et al. [22] also uses diffusion models (DDPM) to augment the training of Lipschitz-based certification methods. Although this work is listed in the references, it is neither discussed nor compared to, despite being the most related approach. This omission leads to several issues:
- The claim in L105 that certified guarantees on ImageNet are “close to random guessing” is inaccurate. Hu et al. report 35% CA for $\ell_2$-norm with $\epsilon = 36/255$, which is significantly higher than the 0.1% of random guessing.
- The claim in L36 that “[generated data] has not yet been combined with deterministic certified training methods” is incorrect, as Hu et al. have done this.
- The claimed improvement over SOTA methods (L13, L145) is not entirely accurate. For $\ell_2$-norm with $\epsilon = 36/255$ in CIFAR-10, Hu et al. report 70.1% CA, which surpasses the 69.05% reported in this work.

**Unclear Meaningfulness of Improvements**

The improvements over prior methods are minor, in the low single digits. As the authors note (L261), a change in hyper-parameters or different seeds can have similar effects on the model. Therefore, it is difficult to judge if the measured improvements are meaningful, especially considering the significant overhead of training a generator, generating the data, and additional model training on the larger dataset. This issue is compounded by the fact that all reported numbers are from single runs with one seed, without error bars or standard deviation. While the authors argue that the large computational cost makes multiple runs infeasible, it remains unclear how meaningful the improvements are.

**Questions:**

L180: Do you really mean “1m, 1m, or 1m auxiliary data”?

**Remarks**

Tables and figures are sometimes far from where they are referenced (e.g., Table 1, Fig 2). This makes it difficult to read as the reader has to jump back and forth.

Some instances of % should be replaced by percentage points, e.g., in the abstract (L14), L42, etc.

**Limitations:**

The limitations are not adequately addressed. The paper should include a discussion of limitations, such as the fact that only four Lipschitz-bound-based certification methods are evaluated. It is unclear if these results generalize to other Lipschitz-based certification methods and are likely not applicable to certification methods from different families. Furthermore, only mid-scale datasets like CIFAR are considered; the generalization gap may not exist in larger-scale datasets. Additionally, the results are limited to image classification, and the resulting improvements are small.

---

> ### Author Rebuttal · Authors · 2024-08-06
>
> Thank you for the feedback, and for highlighting our typo in L180, which was meant to read “1m, 5m, or 10m auxiliary data.” We’d like to address your main concerns as follows.
>
> **Limited Novelty and Insights**
>
> The original idea of using data from generative models trained on the original data actually traces back further to a paper by Gowal et al. [1] in 2021, with Wang et al. [2] later providing a more in-depth empirical analysis of the various factors contributing to the observed improvements. Both publications remain highly influential in the domain of empirical robustness. Although we acknowledge that Hu et al. [3] have now also used DDPM, they only dedicate a single paragraph to its discussion. Hence, our detailed empirical analysis in the domain of Lipschitz-bound certified defenses remains novel. One of the key, and we believe substantial, insights is that additional data can only help close the generalization gap, but not improve robustness itself for Lipschitz-bound certified robustness.
>
> **Missing Comparison to a Key Related Approach**
>
> As outlined previously, Hu et al. [3] might be the first published work to use generative models with Lipschitz-bound certified defenses but is missing a broader set of experiments and discussion of the applicability of data scaling to improve Lipschitz-bound robustness guarantees. We did not include larger-scale datasets, in particular ImageNet, as they are the only ones to achieve meaningful levels of accuracy and also do not apply DDPM. Given the chance, a camera-ready version would better delineate our work from that by Hu et al. [3] as well as reformulate some of the claims mentioned for additional clarity.
>
> **Unclear Meaningfulness of Improvements**
>
> We understand that a larger sample size of different seeds would have been preferable, yet a significant budget of GPU hours (approx. 3000 hours) was already spent on the presented experiments. Given that most trends are present across multiple experiments, we nevertheless feel confident that our conclusions hold up to further scrutiny.
>
> In a camera-ready version we would also revisit placement of tables and figures and ensure that %pt (percentage points) are used in place of % where necessary.
>
> [1] S. Gowal, S. Rebuffi, O. Wiles, F. Stimberg, D. A. Calian, and T. A. Mann, “Improving Robustness using Generated Data,” in Adv. Neural Inf. Process. Syst. (NeurIPS), vol. 34, Virtual Event, Dec. 2021, pp. 4218–4233.
>
> [2] Z. Wang, T. Pang, C. Du, M. Lin, W. Liu, and S. Yan, “Better Diffusion Models Further Improve Adversarial Training,” in Proc. Intl. Conf. Mach. Learn. (ICML), Honolulu, HI, USA, Jul. 2023.
>
> [3] K. Hu, A. Zou, Z. Wang, K. Leino, and M. Fredrikson, “Unlocking Deterministic Robustness Certification on ImageNet,” in Adv. Neural Inf. Process. Syst. (NeurIPS), vol. 36, New Orleans, LA, USA, Dec. 2023.

---

> ### Comment · Reviewer_hXmp · 2024-08-11
>
> Thank you for your response to my questions and comments.
>
> **Limited Novelty and Insights**
>
> As stated in my original review, I agree that the evaluation is more thorough than Hu et al.'s and provides some additional insights. However, the novelty compared to prior work is still limited, as a very similar method has been proposed before.
>
> **Missing Comparison to a Key Related Approach**
>
> Thank you for expanding the discussion and comparison to Hu et al. and for correcting the related claims. The rebuttal alleviates this concern.
>
> **Unclear Meaningfulness of Improvements**
>
> I understand the reason for not running wide-scale additional experiments. However, the small improvements combined with single runs remain a weakness, as it is difficult to judge how meaningful the improvements are. Could it be possible to perform multiple runs with different seeds for some key configurations to limit the computational budget required and still show the mean and variance of results? I don't expect those results within the discussion period, but it would be good to add them to the final version.

---

> > ### Author Response · Authors · 2024-08-11
> >
> > Thank you for your response. We agree that additional experiments with different seeds would help better judge the meaningfulness of the observed improvements and have scheduled four additional runs for the best configurations of each model with and without auxiliary data (e.g., the ones with 64.53% and 69.05% certified accuracies for LOT, Tab. 2). As we are currently running experiments on convex bound propagation we do not expect those results to be ready before the end of the discussion period, though.

---

> ### Author Response · Authors · 2024-08-12
>
> While we are unable to provide true error estimates based on different seeds within the discussion period, we can make an educated guess for GloroNet (L, 2400 epochs) based on a larger-scale sweep with 49 different auxiliary dataset sizes. When fitting a logarithmic curve to the obtained certified accuracies, this yields a standard deviation of 0.20 percentage points under a constant error model (i.e., the error is assumed to be independent of the amount of auxiliary data). This is well below the 2.91 percentage point difference with and without auxiliary data. Although this is certainly limited in methodology, we hope this may help further alleviate any concerns with regards to the significance of the differences observed.

---

### Author Rebuttal · Authors · 2024-08-06

In response to the reviewer’s feedback, we have made the following changes to the manuscript

* In both Sec. 1 (Introduction) and Sec. 7 (Conclusion) we have added further references to Hu et al. and their prior usage of DDPM generated data, and carefully reworded related claims where appropriate (ours remains first for $\ell_\infty$ with new state-of-the-art). This is in response to reviewer #1's valid concerns that we may not have highlighted this enough in our initial manuscript.
* In Sec. 2 (Related Work), subsection “Certified Robustness”, we have dedicated a paragraph to convex bound propagation, and how it differs from Lipschitz-bound certification. This includes the references mentioned by reviewer #3. Experiments are currently running, and we will update once results are in.
* In Sec. 3.1 (Dataset and Threat Models) we have removed the reference to random guessing and replaced it with a more extensive discussion why we chose not to evaluate on the larger-scale ImageNet dataset.
* In Sec. 5 (Comparison to Empirical Robustness), subsection “Amount of training data”, we have added Bayes error as an alternative explanation as to why 1 million additional images seem sufficient for the investigated models. Thanks to reviewer #3 for pointing us in this fairly recently researched direction (May 2024).
* After Sec. 5 we have added a new Sec. 6 (Limitations) which addresses all known limitations, previously scattered throughout the paper, in a single place. Despite our extensive experiments, this includes (a) only evaluating one type of deterministic certified robustness, namely Lipschitz-bound ones; (b) only performing one repetition per experiment; and (c) only work on CIFAR-10 and CIFAR-100, as opposed to ImageNet.

We are currently also running experiments using the work by De Palma et al. [1], which uses convex bound propagation and is the 3rd-best model after ℓ∞-dist Net. However, as we have a shared cluster, we are unable to estimate whether results will be finished by the end of the discussion period. We expect to post an update by Monday.

[1] A. De Palma, R. Bunel, K. Dvijotham, M.P. Kumar, R. Stanforth, and A. Lomuscio, “Expressive Losses for Verified Robustness via Convex Combinations,” in Proc. Intl. Conf. Learn. Representations (ICLR), Austria, AT, May 2024.

---

> ### Author Response · Authors · 2024-08-12
>
> As mentioned by both the first and the third reviewer, our focus on Lipschitz-bound methods raises questions regarding the applicability of data scaling to other certified robustness domains, such as convex bound propagation. Our additional experiments on MTL-IBP [1], a summary of which is reproduced below, indicate vastly different scaling dynamics with no observed improvements even when further adjusting $\alpha$ ($0.05$ being originally used by [1] for the larger ImageNet). A few remarks:
>
> * Verification runs were aborted after 24 hours to accommodate the discussion period. If aborted, reported numbers indicate percentage of certified/correctly classified samples on the subset of CIFAR-10 test images for which verification completed within 24 hours.
> * Non-completed runs included more hard-to-certify samples that required computationally expensive branch-and-bound verification. Extrapolated full runtimes for some configurations are up to about 2 weeks.
> * Runs that yielded 0% certified accuracy oftentimes completed quickly as the comparatively cheap PDG attack already found counterexamples, and thus a full verification was not needed.
>
> We conclude that MTL-IBP, a convex bound propagation approach, cannot benefit from additional generated data. This indicates that results with regards to scalability do not trivially transfer from one robustness method to another, and a detailed investigation of a single method, such as Lipschitz-bound approaches, seems justified.
>
> **Certified Accuracy**
>
> | Model | Epochs | $\alpha$ | None | 50k | 100k | 200k | 500k | 1m | 5m | 10m |
> | ----- | ------ | -------- | ---- | --- | ---- | ---- | ---- | -- | -- | --- |
> | MTL-IBP | 260 | 0.5 | 35.08% | 34.14% | 32.04% | 21.07%* | 0.34%* | † | 0.00% | 0.00% |
> | MTL-IBP | 520 | 0.5 | 35.95% | 34.74% | 34.57% | 33.05% | 17.57%* | 0.54%* | 0.00%* | 0.00% |
> | MTL-IBP | 260 | 0.05 | 28.71%* | 26.67%* | 23.75%* | 8.12%* | 0.00%* | † | 0.00% | 0.00% |
> | MTL-IBP | 520 | 0.05 | 27.76%* | 27.26%* | 26.68%* | 26.72%* | - | - | - | - |
>
> **Clean Accuracy**
>
> | Model | Epochs | $\alpha$ | None | 50k | 100k | 200k | 500k | 1m | 5m | 10m |
> | ----- | ------ | -------- | ---- | --- | ---- | ---- | ---- | -- | -- | --- |
> | MTL-IBP | 260 | 0.5 | 53.69% | 55.46% | 57.70% | 66.04%* | 79.31%* | † | 94.60% | 94.82% |
> | MTL-IBP | 520 | 0.5 | 54.68% | 55.64% | 55.88% | 57.34% | 68.63%* | 79.10%* | 94.60%* | 95.09% |
> | MTL-IBP | 260 | 0.05 | 64.58%* | 66.12%* | 67.67%* | 76.35%* | 84.92%* | † | 94.67% | 94.84% |
> | MTL-IBP | 520 | 0.05 | 65.94%* | 66.08%* | 66.73%* | 67.77%* | - | - | - | - |
>
> \* Verification run did not finish within 24 hours, preliminary result on test data subset shown
>
> † Implementation by [Palma et al.](https://github.com/alessandrodepalma/expressive-losses) triggered an assertion error
>
> [1] A. De Palma, R. Bunel, K. Dvijotham, M.P. Kumar, R. Stanforth, and A. Lomuscio, “Expressive Losses for Verified Robustness via Convex Combinations,” in Proc. Intl. Conf. Learn. Representations (ICLR), Vienna, AT, May 2024.

---

### Decision · Program_Chairs · 2024-09-25

**Decision:**

Accept (poster)

**Comment:**

This paper proposes to use diffusion models for data augmentation to improve the certified robustness of image classification models. The paper is well-written and easy to follow, and the the proposed method achieves state-of-the-art results on CIFAR-10, demonstrating a significant improvement in robustness certificates for $\ell_\infinity$ and $\ell_2$ perturbations. I suggest the authors add the additional experimental results mentioned in the rebuttal into the final version. I recommend accept (poster).